# LANGUAGE-GUIDED OBJECT-CENTRIC WORLD MODELS FOR PREDICTIVE CONTROL

## ABSTRACT

A world model is essential for an agent to predict the future and plan in domains such as autonomous driving and robotics. To achieve this, recent advancements have focused on video generation, which has gained significant attention due to the impressive success of diffusion models. However, these models require substantial computational resources. To address these challenges, we propose a world model leveraging object-centric representation space using slot attention, guided by language instructions. Our model perceives the current state as an object-centric representation and predicts future states in this representation space conditioned on natural language instructions. This approach results in a more compact and computationally efficient model compared to diffusion-based generative alternatives. Furthermore, it flexibly predicts future states based on language instructions, and offers a significant advantage in manipulation tasks where object recognition is crucial. In this paper, we demonstrate that our latent predictive world model surpasses generative world models in visuo-linguo-motor control tasks, achieving superior sample and computation efficiency. We also investigate the generalization performance of the proposed method and explore various strategies for predicting actions using object-centric representations.

## 1 INTRODUCTION

A world model, or world simulator, enables an agent to perceive the current environment and predict future environmental states. By providing the agent with future environment states corresponding to the actions it can take, it helps planning and policy learning in control tasks, e.g., autonomous driving (Hu et al., 2022; Pan et al., 2022; Hu et al., 2023; Zhang et al., 2023; Wang et al., 2023), gaming (Schrittwieser et al., 2020; Hafner et al., 2019; 2020; 2023), and robotics (Black et al., 2023; Du et al., 2024a;b; Yang et al., 2024; Zhou et al., 2024).

With the remarkable success of diffusion models, there has been a growing interest in employing video-generation-based world models, particularly those that are conditioned on the current frame and language instructions, to perform planning and control tasks (Du et al., 2024a;b; Yang et al., 2024).

However, the major drawback of language-guided video-generation models is the requirement of large-scale labeled language-video datasets and the corresponding high computational cost (Gu et al., 2024). Therefore, latent predictive models, which abstract video to predict forward in compact latent state spaces, can serve as an alternative from a computational efficiency perspective (Hafner et al., 2019). The key points of these models are how to abstract the video and how to predict the future state, and various studies have explored these strategies (Schrittwieser et al., 2020; Hafner et al., 2019; 2020; 2023; Seo et al., 2023).

Meanwhile, object-centric representation derived from Locatello et al. (2020) has recently garnered significant attention as a method for encoding images or videos in several studies (Kipf et al., 2021; Elsayed et al., 2022; Seitzer et al., 2023; Bao et al., 2023; Singh et al., 2022a;b; Aydemir et al., 2024; Zadaianchuk et al., 2023). In this approach, the representation is primarily learned through an auto-encoding structure that reconstructs frames or pretrained feature of the frames, and it has been reported that the learned representation not only aids in reconstruction but also benefits control tasks (Heravi et al., 2023; Yoon et al., 2023; Driess et al., 2023).

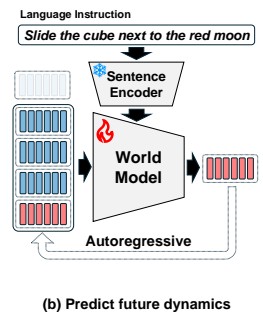

Figure 1: Training and inferencing overview of our world model. (a) During training, frames are processed through a pre-trained slot encoder to extract slots, and language instructions are processed using a sentence encoder. Slots along with the instruction representation, are used to condition the world model, which predicts the slots for future states. These predicted slots are then compared with the future ground truth slots extracted from the frames, and a reconstruction loss is computed to train the world model. (b) More specifically, the world model utilizes the predicted slots from the previous steps to autoregressively predict future slots.

Inspired by this, we aim to fully exploit the advantages of object-centric representation in world modeling and control in this work. We propose an object-centric world model that uses a slot-attention-based encoder to obtain object-centric representations from observations and then predicts the representations of future states conditioned on given language instructions. Through this approach, we can represent observations compactly while enabling expressive and flexible predictions guided by language instructions. Additionally, providing imagination in an object-centric form is beneficial for control tasks where object recognition is essential. We evaluate this approach on visuo-linguo-motor control tasks to highlight the sample and computational efficiency compared to the approach with a state-of-the-art diffusion-based generative model.

In summary, the contributions of our work are as follows:

- To the best of our knowledge, we are the first to propose object-centric world models guided by language instruction.
- We show our approach surpasses the alternative using state-of-the-art diffusion-based generative model in both visuo-linguo-motor control task success rate and computational speed, highlighting our method's sample and computation efficiency.
- We explore the generalization performance of the object-centric world model guided by language instruction on unseen task settings.
- We explore various methods for predicting actions using slots, a topic addressed in only a few studies.

## 2 RELATED WORKS

### 2.1 OBJECT-CENTRIC REPRESENTATION LEARNING

Object-centric representation learning is a method that decomposes objects in an image and binds them into a structured latent spaces, called slot, without any supervision (Greff et al., 2019; Burgess et al., 2019; Engelcke et al., 2019; Locatello et al., 2020). This approach demonstrates its efficiency and performance in various tasks such as object discovery (Biza et al., 2023; Fan et al., 2024) and segmentation (Stelzner et al., 2021; Xu et al., 2022; 2023), where clearly distinguishing the objects is essential. In recent years, many studies have focused on applying object-centric approaches to video (Bao et al., 2022; Lee et al., 2024; Aydemir et al., 2024). Kipf et al. (2021) enhances video understanding by extracting slots using spatio-temporal slot attention, which applies slot attention to each frame and each slot individually. Additionally, research based on object-centric approaches is being conducted in the field of video prediction tasks. Wu et al. (2022) proposes a method for

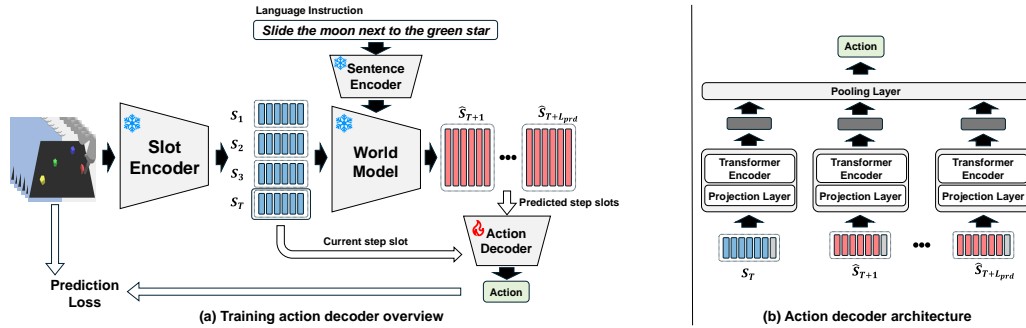

Figure 2: Overview of the action decoder training process. (a) The action decoder is trained by inputting the current state slots and future state slots obtained from the trained world model to predict actions. (b) The detailed architecture of the action decoder is as follows: input slots are grouped by timestep and passed through a projection layer with shared weights, followed by a transformer encoder. The outputs are then concatenated in chronological order and fed into a pooling layer to predict the action.

predicting the future state of objects in an autoregressive manner using a transformer model that understands visual dynamics by leveraging slots extracted through a slot encoder. In this study, we introduce a language-based model built on the Slotformer architecture, which predicts actions by fusing slots extracted from SAVi with instructions. To the best of our knowledge, this is the a first language-guided world model using an object-centric representation.

## 2.2 WORLD MODEL

A world model predicts future states based on current environments. Recent advancements in diffusion models have led to active research on video generation-based world models (Du et al., 2024a;b; Yang et al., 2024), but these methods demand substantial data and lengthy training and inference times. Some studies address these challenges by learning dynamics within the latent space, either by extracting full-image representations (Hafner et al., 2019; 2020; 2023; Babaeizadeh et al., 2017; Franceschi et al., 2020) or by using masked inputs (Seo et al., 2023). Others propose object-centric approaches, focusing on state representations (Wu et al., 2022; Collu et al., 2024) or combining states and actions (Ferraro et al., 2023; Feng & Magliacane, 2023).

We introduce the first object-centric world model guided by natural language. Our model is more computationally efficient than diffusion-based approaches and outperforms non-object-centric methods in action prediction tasks. Unlike prior studies that train goal-conditioned policies using object-centric representations requiring goal images at test time (Zadaianchuk et al., 2020; Haramati et al., 2024), our approach predicts future states based on instructions and uses these predictions to train an action decoder, enabling manipulation tasks without goal images.

## 2.3 VISUO-LINGUO-MOTOR CONTROL TASKS

This task is designed to evaluate how effectively an agent can control given situations based on visual perception and linguistic understanding (Shridhar et al., 2023; Wang et al., 2024; Kim et al., 2024). Specific tasks are provided in the form of instructions, along with visually recognizable environments. We test our world model's visuo-linguo-motor control capabilities using the language-table dataset (Lynch et al., 2023). It consists of a simulation environment with a single robot arm and four blocks, where visual data is provided in the form of videos, and linguistic information is given in instructions such as 'put the blue block next to the pentagon.'

## 3 METHODOLOGY

In this section, we propose methods and components for our language-guided world model on object-centric representation space and predictive control.

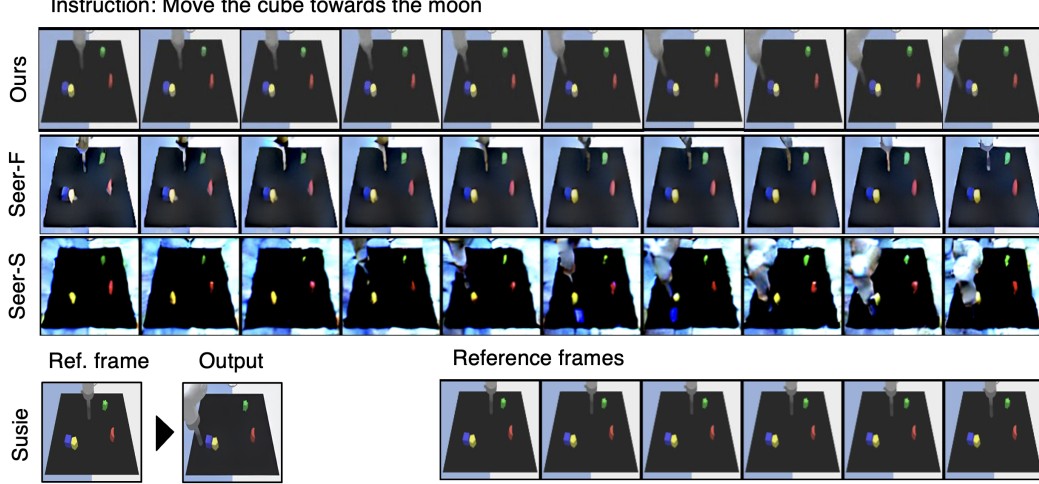

Figure 3: Decoded video frames of our method, Seer, and decoded frame of Susie, conditioned on given reference frames and the instruction, 'Move the cube towards the moon.' Seer-F produces higher quality generations compared to Seer-S, but both methods fail to predict states guided by the instruction. Susie successfully generates the future frame conditioned on the reference frame and the instruction. Seer results are generated using 30 DDIM sampler steps and Susie uses 10 steps.

## 3.1 LANGUAGE-GUIDED SLOT WORLD MODEL

Given a video history $\boldsymbol{X}_{\mathrm{prv}} = [\boldsymbol{x}_1, ..., \boldsymbol{x}_T] \in \mathbb{R}^{T \times C \times H \times W}$ of previous image observations at time $T$, we first extract object-centric slots, $\boldsymbol{S}_{\mathrm{prv}} = f_{\mathrm{enc}}(\boldsymbol{X}_{\mathrm{prv}}) \in \mathbb{R}^{T \times K \times D}$, where $K$ is a pre-defined number of slots and $D$ is the dimension of a slot embedding. We use SAVi (Kipf et al., 2021) as a video slot extractor, which iteratively applies slot attention along the temporal axis to update and refine the slots that represent distinct entities within the video frames. This slot extractor is pre-trained on the target domain before training the world model. All following experiments are executed in slot latent space and the trained SAVi decoder is only utilized when visualizing extracted or predicted slot representations in image space.

From the extracted slots $\boldsymbol{S}_{\mathrm{prv}}$ and a text instruction $I \in \mathbb{R}^{D_{\mathrm{txt}}}$, the world model autoregressively predict future slot trajectory for rollout prediction steps $L_{\mathrm{prd}}$ via prediction step $\hat{\boldsymbol{s}}_{T+t} = f_{\mathrm{wm}}([\boldsymbol{S}_{\mathrm{prv}}, \boldsymbol{S}_{\mathrm{prd}}]_{\geq T+t-L_{\mathrm{cnd}}} | I)$ where $\boldsymbol{S}_{\mathrm{prd}} = [\hat{\boldsymbol{s}}_{T+1}, ..., \hat{\boldsymbol{s}}_{T+L_{\mathrm{prd}}}] \in \mathbb{R}^{L_{\mathrm{prd}} \times K \times D}$ is an accumulated future predicted slots and $L_{\mathrm{cnd}}$ is conditioned frame length of the world model.

Prior work SlotFormer (Wu et al., 2022), which uses a transformer-based model to autoregressively predict future trajectory given past observations in object-centric representation space, shows that it can capture spatio-temporal object relationships, accurately predict the future, and serve as a world model for downstream tasks.

For our language-guided world model, we propose LSlotFormer, a modified version of the SlotFormer with the language instruction embedding $I$ from T5-base sentence encoder (Raffel et al., 2020) to be integrated as context of transformer decoder. While SlotFormer unconditionally predicts next frames under natural flow of the scene, our model has control over the object dynamics to predict trajectory following an instruction by language guidance. As shown in (b) of Figure 1, LSlotFormer autoregressively predicts future slots conditioned by the language instruction, which is trained to reconstruct the slot representation of target ground truth video $\boldsymbol{S}_{gt} = f_{\mathrm{enc}}([\boldsymbol{X}_{\mathrm{prv}}, \boldsymbol{X}_{gt}])$ for every timesteps $t \in [1, L_{\mathrm{prd}}]$ and slots $k \in [1, K]$ with $\mathcal{L}_{\mathrm{slot}}$.

$$\mathcal{L}_{\mathrm{slot}} = \frac{1}{L_{\mathrm{prd}}} \frac{1}{K} \sum_{t=T+1}^{T+L_{\mathrm{prd}}} \sum_{k=1}^{K} ||\hat{s}_{tk} - s_{tk}||^2 \tag{1}$$

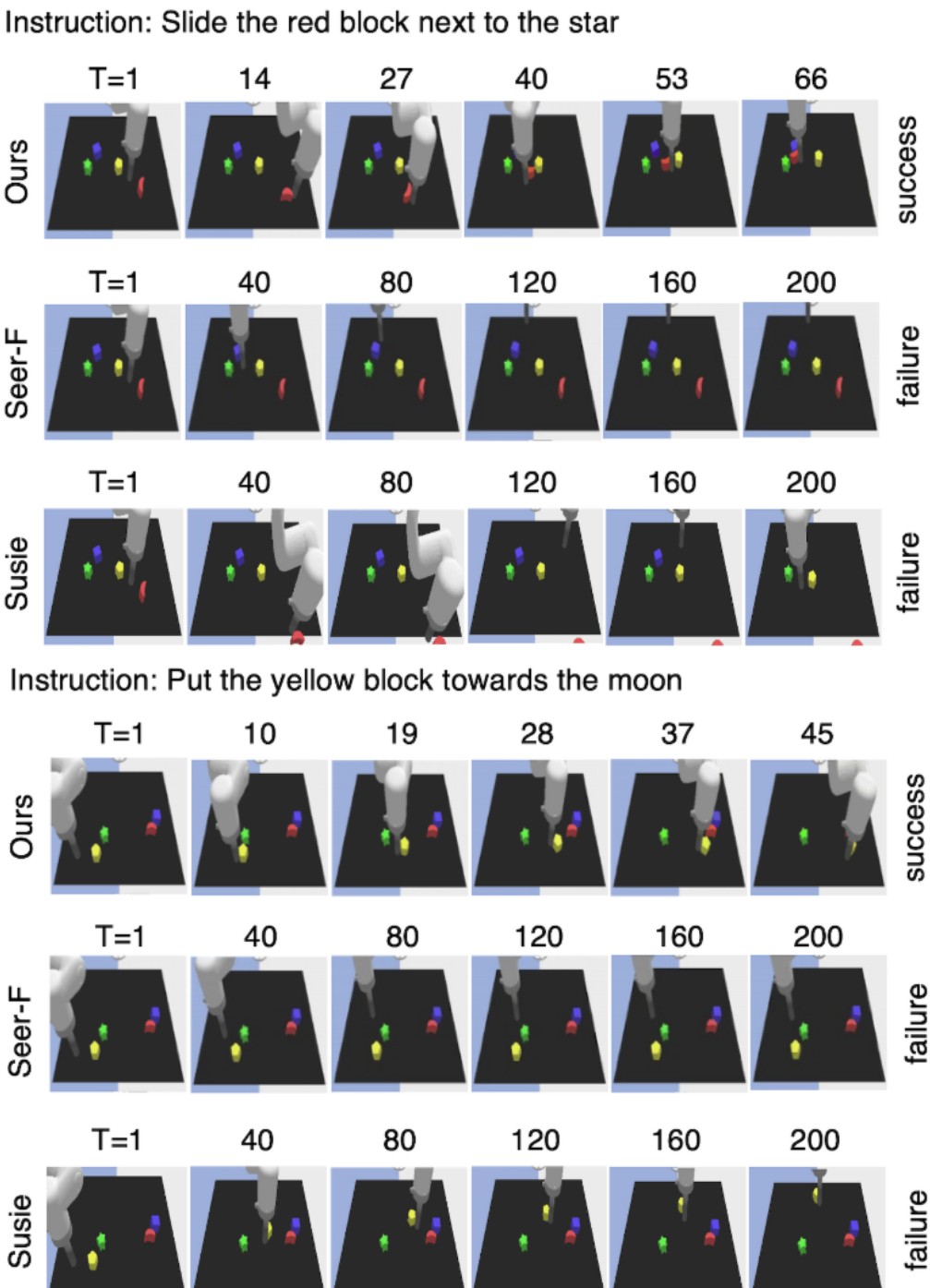

Figure 4: Predicted action trajectory of ours and the baseline in visuo-linguo-motor control simulation environment. When given the instructions, our method successfully recognizes and moves the correct blocks to complete the episodes, while the baselines fail.

## 3.2 PREDICTIVE CONTROL

With current slots $s_T$, predicted future slots $S_{\text{prd}}$ from the world model, and an optional language instruction $I$, the objective is for the action decoder network to predict agent action $\hat{a}_T =$

Table 1: Control task results. The results of each model in seen tasks and blocks are presented with task success percentages based on the threshold distance of success. The results of each model in unseen tasks and blocks are presented with the threshold distance 0.05.

| Model | Seen tasks & blocks | | | Unseen tasks | | | Unseen blocks |
|---|---|---|---|---|---|---|---|
| | 0.05 | 0.075 | 0.1 | T1 | T2 | T3 | |
| Seer-S + SAVi | 0.0 | 1.0 | 0.0 | 0.0 | 0.0 | 0.0 | 0.5 |
| Seer-F + SAVi | 0.5 | 0.0 | 1.0 | 0.0 | 0.0 | 0.5 | 1.0 |
| Seer-S + VAE | 0.5 | 0.0 | 0.0 | 0.0 | 0.0 | 0.0 | 0.0 |
| Seer-F + VAE | 0.5 | 1.0 | 0.5 | 0.5 | 0.0 | 0.5 | 0.5 |
| Susie + SAVi | 13.0 | 18.5 | 23.0 | 11.0 | 10.5 | 8.0 | 15.5 |
| Susie + VAE | 12.0 | 14.0 | 20.5 | 13.5 | 9.5 | 9.0 | 14.0 |
| Ours | **50.0** | **61.5** | **73.0** | **24.5** | **16.0** | **26.5** | **38.0** |

Table 2: Comparison of our method and baselines in terms of dataset, video quality and computational speed.

| Model | Training Data | Number of Data | Video Generation | Speed | |
|---|---|---|---|---|---|
| | | | FVD | Training (s/it) | Inference (s/it) |
| Seer | Language-Table | 8,000 | 641.84 | 0.40 | 0.72 |
| Seer | Language-Table + Something-Something V2 | 220,847 | **205.94** | | |
| Susie | Language-Table + InstructPix2Pix | 459,990 | - | 0.18 | 0.47 |
| Ours | Language-Table | 8,000 | 346.59 | **0.06** | **0.19** |

$\pi(s_T, S_{\mathrm{prd}}, I)$. This inverse dynamics policy is trained by supervised behavioral cloning, training the agent to follow the ground truth trajectory with action loss $\mathcal{L}_\pi$.

$$\mathcal{L}_\pi = ||\pi(s_T, S_{\mathrm{prd}}, I) - a_T||^2 \tag{2}$$

Beyond slots extraction and reconstruction, it is relatively unexplored of how to deal with slot features in downstream tasks with proper inductive bias. Therefore, we explore various action decoding mechanisms depending on which architectures to use, whether to use language instruction, and how to merge features from individual slots. Details of this exploration of variations are explained in ablations. As a result of this experiments, we use transformer layers to decode slots to an action feature for every timestep and fuse the concatenated action features with MLP pooling layers to an action prediction without instruction being used. This process is depicted in (b) in Figure 2 and is used as the base setting for our experiments unless mentioned otherwise.

## 4 EXPERIMENTS AND RESULTS

### 4.1 DATASETS

We train and evaluate our methods on language-table (Lynch et al., 2023), a simulated environment and dataset for language-guided robotic manipulation. Specifically, a block-to-block task subset with 4 blocks containing 8,000 human-conducted trajectories where 7,000 trajectories are split for training is used. Then trajectories filtered with a minimum length of 50 are utilized for our experiments. The task is to move a block to another block based on the description of the colors or shapes

in a scene consisting of a fixed combination of block colors and shapes, which are the red moon, blue cube, green star, and yellow pentagon.

## 4.2 BASELINES

To compare our approach with diffusion-based generative world models, we employ Seer (Gu et al., 2024), one of the state-of-the-art language-guided video generation models based on a latent diffusion model, and Susie (Black et al., 2024), which generates a future state image conditioned on language instruction and current state image using fine-tuned InstructPix2Pix (Brooks et al., 2023) as the world model.

We train Seer using two approaches: training it from scratch on the language-table dataset (denoted as Seer-S) and fine-tuning it on the language-table dataset from a checkpoint pre-trained on the Something-Something V2 dataset (Goyal et al., 2017) (denoted as Seer-F). Additionally, we utilize the latent features from baselines in two different ways. The first method involves inputting the latent $z_0$, obtained before reconstruction into the video space through the VAE decoder in the latent diffusion model, into a CNN-based action decoder to output actions (denoted as +VAE). The second method involves extracting slots from the reconstructed video using the same SAVi model employed in our approach, and then inputting these slots into an action decoder with the same architecture as our approach to output actions (denoted as +SAVi). This approach allows us to better isolate the role of the world model in the control task, as both ours and the baseline use the same SAVi encoder and action decoder. Similar to Seer, Susie utilizes a VAE and SAVi model as encoders to decode actions from latent features. In Susie, we employ an action decoder with the same structure as Seer, except for the input feature and the output action dimension. It only uses features of two images as input and predicts an action trajectory between the current and subgoal images, rather than a single action.

## 4.3 EVALUATION SETUP

We compare the success rates of each approach by randomly sampling 200 episodes in the language-table environment. Each episode is set up similarly to those used during the training of the language-guided world model and action decoder, involving four blocks and an episode to move one block to another block. For each sampled episode, the positions of the blocks, the block to be moved, and the target block are randomly determined. In the environment, an episode is deemed successful if the block to be moved comes within 0.05 unit distances of the target block. We evaluate this success criterion not only at the 0.05 threshold but also at additional unit distances (i.e., 0.075 and 0.1). As the distances increase, it becomes easier to complete the episode. We consider an episode successful if it is completed within 200 steps. The DDIM sampler for the baselines is set to 10 steps, a value determined through trial and error to achieve high-quality generation.

## 4.4 CONTROL USING WORLD MODEL

**Control task** First, we compare the control task performance of our approach with that of the baselines. We report the task success rate of 200 randomly sampled episodes from the language-table environment, using the same setup as the training data.

Our approach consistently outperforms all variants of Seer in terms of success rate across all three target distances (success criteria), as shown in Table 1. Whether trained from scratch using the same amount of video data as our method or fine-tuned from a pre-trained checkpoint, Seer exhibits a consistently lower success rate compared to our approach, underscoring the superior sample efficiency of our method. This outcome likely stems from the world model component playing a more critical role than the action decoder. As illustrated in Figure 4, Seer consistently fails to accurately predict the arm's movements and the resulting future states of the blocks necessary to complete the episodes. This is further evidenced by the fact that even when using the frozen SAVi encoder from our approach to predict actions based on the future videos generated by the baseline, the success rate remains lower than that of our method.

Next, we perform a qualitative analysis through visualization. In Figure 3, we present 10 future frames decoded from the latent predictions of each world model after observing the given reference frames. Our decoded frames accurately depict the robot arm moving toward the cube, to execute the given instruction 'move the cube towards the moon.' The Seer model trained from scratch retains

some object details. The fine-tuned Seer model produces frames more consistent with better quality, yet neither Seer model successfully predicts arm movements that reflect an understanding of the instruction. Considering that both our approach and Seer use a frozen image decoder, Seer fails to predict future latent states with language instruction guidance.

Meanwhile, Susie shows better predictions of the arm's movement and block states compared to Seer (Figure 3), but the action trajectory predicted by its action decoder is inaccurate (Figure 4). This suggests that a temporally coarse world model may accurately predict states, but it struggles to accurately predict long low-level action trajectories.

**Computation speed**   We report the computation speed of world models during both training and inference. Inference refers to evaluation within the language-table environment using the world model. Training is evaluated with a batch size of 8 using 4 H100s, while inference is evaluated with a single episode, using a single H100.

As shown in Table 2, ours is faster than the baseline in both training and inference. Our approach completes training in up to 85% less time (Seer: 0.40s/it, Susie: 0.18s/it, Ours: 0.06s/it) and inference in up to 74% less time (Seer: 0.72s/it, Susie: 0.47s/it, Ours: 0.19s/it) compared to the baselines. Note that this comparison is based on the DDIM sampler with 10 steps. Increasing the number of steps for higher-quality generation could further widen the computation speed difference.

### 4.5 GENERALIZATION

**Unseen blocks**   We test our model's ability to generalize to different types of blocks. Since the dataset only contains four fixed color-shape combinations of the blocks, the experiments are conducted on settings where two of the color-shape combinations of the blocks are changed. It is done by swapping the shapes to avoid ambiguity of the instruction, for example, green pentagon and yellow star instead of green star and yellow pentagon. The results when changing two blocks is presented in Table 1, highlighting the generalization capability of each methods.

Table 3: Action decoder ablation results. Design choices of action decoder presented with their task success percentage by the threshold distance of success.

| Method | Instruction | Action decoder | | Success rate | | | |
|---|---|---|---|---|---|---|---|
| | | Transformer | MLP | 0.05 | 0.075 | 0.1 | Mean |
| Slot group | ✓ | ✓ | | 43.0 | 57.0 | 63.5 | 54.5 |
| | | ✓ | | 43.5 | 58.5 | 64.0 | 55.3 |
| Time group | ✓ | | ✓ | 19.0 | 22.5 | 32.5 | 24.7 |
| | | | ✓ | 20.5 | 30.0 | 40.5 | 30.3 |
| | ✓ | ✓ | | 46.0 | 59.5 | 69.5 | 58.3 |
| | | | ✓ | **50.0** | **61.5** | **73.0** | **61.5** |

Table 4: Success rate based on the number of future steps provided to the action decoder. Note that 0 future steps mean providing the action decoder with the current state slots and the instruction, whereas in the other cases, only the current state and future state slots are provided.

| Method | Future steps | | | | |
|---|---|---|---|---|---|
| | 0 | 1 | 5 | 10 | 20 |
| Ours | 2.5 | 33.5 | 47.0 | **50.0** | 49.0 |

**Unseen tasks**   We further evaluate our method's generalization to unseen tasks. This includes other task configurations provided by language-table: (T1) *Block to Purple Pole*, where a block should be moved to an unseen purple pole object; (T2) *Block to Absolute Position*, where a block should be moved to an absolute position of the table; (T3) *Block to Block Relative Position* where a block should be moved to a relative position of another block. The results of unseen tasks are presented in Table 1 with performances leaving room for further enhancement of generalization. We note that the instructions for these tasks include words that our world model had not previously encountered during training.

## 4.6 ABLATIONS ON ACTION DECODING

**Is a world model really necessary?** We experiment to answer the following question: is a world model really necessary for predicting future actions? Some studies (Lynch et al., 2023; Heravi et al., 2023) only use the current scene and the task information to predict actions to solve the task. We compare the performance of a model that only uses the current state slots and instruction with models that use 1, 5, 10 or 20 future state slots without instruction in Table 4. We show that the model with no access to future state slots almost entirely fails to complete the tasks, while the performance significantly improves as soon as it is given access to even a single future state slot. To investigate whether the issue could simply be a lack of sufficient state information, we also evaluate a model that is provided with the current and past 5 steps of state information. This model also mostly fails (with a success rate ranging from 0.5% to 1.0%), indicating that successful visuo-linguo-motor control tasks require the model to access future state predictions through a world model.

**How far should the model look?** We experiment to find the appropriate number of next steps that the world model should provide to the action decoder for accurate action prediction. As demonstrated in Table 4, This shows that at least one next step must be provided for the action decoder to predict actions. Additionally, the best performance is achieved when 10 steps are given, and providing more future slots does not further improve the accuracy of action prediction. This reveals that information about the distant future is not helpful when predicting actions for the near future, unless modeled with specific methodologies like goal conditioning for long-term decision making.

**Does past information help the model?** We have previously demonstrated that providing information from 10 future frames yields the best control performance. This leads us to the next question: Would adding past frame information to this setup further enhance performance? We compare the performance of a model that uses current state slots and the 10 future state slots with models that use the 5 past state slots, current state slots, and the 10 future state slots. The former shows a success rate ranging from 50.0% to 73.0%, while the latter demonstrates a success rate between 45.5% and 66.5%. The model's ability to predict or control based on future frames appears to be more influential, suggesting that access to upcoming states plays a more crucial role in achieving better performance than integrating past state information.

**Does adding instructions into the action decoding help?** In our approach, we fuse language instructions into the slots within the world model to obtain future state slots, which are then used to predict actions. This raises the following question: Would incorporating the instruction directly into the action prediction process also be beneficial? To answer this question, we modify our approach by concatenating the language instruction, passed through a T5 encoder as a single token, with the slots input to the action decoder. This combined input is then used to decode the action. As shown in Table 3, adding the instruction to the action decoding process lowers performance. This suggests that the predicted slots already contain the necessary instruction information, and reintroducing the instruction during action decoding may interfere with the process. We present additional experiments about fusing instruction with slots to decode actions in the Appendix.

**How to decode actions using slots?** Most research related to slots has primarily focused on object segmentation. While it is known that using slots can be beneficial for control tasks, little is known about how to use slots specifically for action prediction. Yoon et al. (2023) discusses the design choices for action decoders when using slots in reinforcement learning tasks, showing that transformers outperform simple MLP pooling layers. They attribute this to the permutation invariance of slots and the superior performance of transformers.

We compare the choice of MLP versus transformer encoder for action decoding. MLP directly predicts actions with flattened inputs through all timesteps, using ResNet MLP proposed in Lynch et al. (2023). We compare two options using a transformer encoder, grouping by timestep (ours) and grouping by slot. Grouping by timestep treats each slot in the same timestep as a single token and inputs them into a transformer encoder. The outputs from all timesteps are then concatenated chronologically and passed through an MLP layer to predict the action. Grouping by slot treats an individual slot across all timesteps as a single input and passes it through a transformer layer. The reason for using a transformer instead of an MLP for pooling is that when grouping by slots and processing each slot separately, we expect the output to also exhibit permutation invariance.

Generally, we find that the transformer encoder is better in success rate than MLP. This indicates that handling slots across multiple timesteps requires a more sophisticated structure than a simple MLP. For two transformer encoder options, grouping by timestep (ours) results in a 6.5%p to 9%p higher success rate, depending on the success criteria. This reveals that when using multiple slots from multiple timesteps to predict actions, grouping by slot is able to capture useful information from slots for actions.

### 4.7 Decoded Video Quality

We report the Frechet Video Distance (FVD) of our method and Seer in Table 2, calculated using the Kinetics-400 pre-trained I3D model (Carreira & Zisserman, 2017). FVD is evaluated on 130 samples from the validation sets. For FVD, we adhere to the evaluation code provided by Gu et al. (2024).

In terms of FVD, the Seer-F scores the best at 205.94, followed by our approach at 346.59, while the Seer-S has the worst score at 641.84. Our approach outperforms Seer-S on both metrics and shows only a small difference compared to Seer-F, which is trained on significantly more data. This further highlights the sample efficiency of our method.

## 5 Conclusion

In this work, we propose a language-guided world model in object-centric representation space and leverage its predictive capabilities to visuo-linguo-motor control tasks. Our method exceeds using state-of-the-art diffusion-based generative model as a world model on both performance and efficiency for future state prediction and control tasks. Furthermore, we conduct evaluations on unseen blocks and task settings presenting the generalization performance of our method. Our work shows the potential of language-guided object-centric world models to benefit robotics perception and control.

**Limitations** One of our work's limitations is that it only covers data domains from simulated environments. It can be extended to complex real-world videos using a pre-trained vision model like ViT as the primary encoder for slot extraction like Seitzer et al. (2023). Moreover, improved data diversity and techniques (Fan et al., 2024) can be deployed to overcome limitations of generalization performance and the nature of slot attention not being robust to variable object numbers. Also, the deterministic nature of SAVi with learnable slot initialization and SlotFormer remains, leaving future works for modeling stochasticity in the world model.

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

# A   APPENDIX

## A.1   IMPLEMENTATION DETAILS

**SAVi**   Our implementation is based on the official SlotFormer (Wu et al., 2022) codebase[1]. Following $128 \times 128$ resolution settings, our model is trained with a slot embedding dimension of 128 and learnable slot initialization. The number of slots is chosen as 6 considering 4 blocks, 1 arm, and background. An image from a single time step is encoded using CNN layers as Table 5. Then slot attention iterates two times with the slots as a query and the image features as key and value. The resulting slots are fed to the slot predictor, which is 2 layers of transformer encoders with 4 heads followed by LSTM cells with a hidden size of 256. The slot predictor models slot interactions and scene dynamics and outputs slots for the next time step. This process is iterated again temporally with a clipped trajectory with a length of 6. While SAVi can be conditioned with prior properties to initialize the slots or use optical flow as a training objective, we stick to the setting of unconditional reconstruction of an original image to remove the need for additional supervision or estimation networks.

Table 5: SAVi encoder architecture

| Layer | Stride | # Channels | Activation |
|-------|--------|-----------|------------|
| Conv 5x5 | 2×2 | 64 | ReLU |
| Conv 5x5 | 1×1 | 64 | ReLU |
| Conv 5x5 | 1×1 | 64 | ReLU |
| Conv 5x5 | 1×1 | 64 | ReLU |
| PosEmb | - | 64 | - |
| LayerNorm | - | - | - |
| Conv 1x1 | 1×1 | 128 | ReLU |
| Conv 1x1 | 1×1 | 128 | - |

Table 6: SAVi decoder architecture

| Layer | Stride | # Channels | Activation |
|-------|--------|-----------|------------|
| Spatial Broadcast 16x16 | - | 128 | - |
| PosEmb | - | 128 | - |
| ConvTranspose 5x5 | 2×2 | 64 | ReLU |
| ConvTranspose 5x5 | 2×2 | 64 | ReLU |
| ConvTranspose 5x5 | 2×2 | 64 | ReLU |
| ConvTranspose 5x5 | 2×2 | 64 | - |
| Conv 1x1 | 1×1 | 4 | - |

For image reconstruction, each slot is decoded with shared-weight spatial broadcast decoder (Watters et al., 2019) of which architecture is shown in Table 6. Slot-wise outputs have 4 channels representing a reconstructed image and a mask, which are combined to a full image reconstruction via weighted sum using the softmax normalized masks.

**LSlotFormer**   Our LSlotFormer is also based on the official SlotFormer (Wu et al., 2022) codebase. It uses BERT (Devlin et al., 2019) with Pre-LN Transformer (Xiong et al., 2020) design. We use 8 layers of transformer decoders with 8 heads and 256 hidden sizes. Slots are processed with sinusoidal positional encoding only in the temporal axis to keep the permutation equivariance of the slots. Projected instruction embedding is injected as a context to the cross attention in the transformer decoder. We autoregressively rollout 10 frames with slots from the past 6 frames as input.

**Action decoder**   Our action decoder is based on the transformer encoder. The transformer encoder processes the slots of a single timestep as a sequence and then concatenates a learnable action token

---

[1]https://github.com/pairlab/SlotFormer

to generate the output for that timestep. This process is repeated for all timesteps, and the outputs obtained from each timestep are concatenated and fed into an MLP layer to produce the final action. Each token has a dimension of 128, and we use 8 heads and 2 layers. The dimension of the feed-forward network is 128, and no separate positional encoding is applied to the slots. The final MLP layer consists of a single layer that directly outputs the action.

**Seer**  We use two variants of Seer. Seer-S is fine-tuned on the language-table dataset for 200,000 steps based on Stable Diffusion V1.5, while Seer-F is initialized from a checkpoint fine-tuned on the Something-Something V2 dataset for 200,000 steps and further fine-tuned on the language-table dataset for an additional 20,000 steps. Both variants are trained to generate images at a resolution of (128, 128), conditioned on 6 frames to generate 10 frames. Seer-S is trained with a batch size of 8 and a learning rate of 2.56e-6, with a learning rate warmup of over 10,000 steps. Seer-F uses the same batch size but a learning rate of 2.56e-7, with a learning rate warmup over 1,000 steps. Both variants utilize the AdamW optimizer with $\beta_1$ of 0.9, $\beta_2$ of 0.999, a weight decay of 0.01, and epsilon set to 1e-8. A cosine learning rate scheduler is employed.

The SAVi encoder and the corresponding action decoder are the same as those used in our model. The structure of the action decoder that processes the VAE features is as follows. First, CNN is constructed as a sequential neural network that starts with a 2D convolutional layer with 4 input channels and 32 output channels, using a kernel size of 3, a stride of 1, and padding of 1. This is followed by a ReLU activation function to introduce non-linearity. Next, a 2D max-pooling layer with a kernel size of 2 and a stride of 2 is applied for down-sampling. The sequence continues with a second 2D convolutional layer, which takes 32 input channels and outputs 64 channels, again using a kernel size of 3, a stride of 1, and padding of 1. Another ReLU activation function is applied, followed by a second 2D max-pooling layer with the same kernel size and stride as before. Afterward, the output of the CNN is flattened and passed through an MLP that maps it to a 128-dimensional vector. This is followed by a ReLU activation layer, and finally, another MLP is applied to predict the action.

**Susie**  We initialize Susie with a checkpoint from InstructPix2Pix, which was trained on 451,990 data samples, and fine-tune it on the language-table dataset for 40,000 steps. The model is trained with a batch size of 128 and a learning rate of 1e-4, with a warmup period of 800 steps. We use the AdamW optimizer with $\beta_1$ of 0.9, $\beta_2$ of 0.999, epsilon set to 1e-8, and a weight decay of 0.01. A cosine learning rate scheduler is applied. The model is trained to generate images with a resolution of (256, 256), and all other settings are kept identical to those in the official codebase configurations.

Susie's SAVi and VAE encoders have the same settings as those in Seer, and while the overall structure of the action decoder is also similar, it differs in two key aspects: it takes as input the features of two images (the current and subgoal), and instead of outputting a single action, it predicts an action trajectory.

**Experiment Configurations**  Training configurations are stated in Table 7. We note that past frames and rollout frames can differ depending on ablation experiments. Our method is trained in a single node environment with 4 RTX 3090s and diffusion baselines are trained with 4 H100s. Computation speed experiments comparing the methods are conducted with 4 H100s for training and 1 H100 for inference.

## A.2  ADDITIONAL EXPERIMENT RESULTS

**Masking slots in action decoder**  We hypothesize that merging these separated object features may negatively impact performance. To ensure that the separated object information in each slot is preserved during action decoding, we conduct additional experiments by applying a mask when inputting the slots into the transformer. With the mask, each slot can only attend to itself, while the action token attends to all slots. The experimental results, presented in Table 8, show that using the mask generally leads to performance degradation across almost all scenarios. This suggests that allowing the slots to mix during action prediction results in better performance.

**How to fuse instructions into slots in action decoder?**  To explore alternative methods of fusing instructions with slots in the action decoder, we conduct additional experiments by using a trans-

Table 7: Training configurations

| Config | SAVi | LSlotFormer | Action Decoder |
|---|---|---|---|
| Epochs | 40 | 40 | 10 |
| Batch size | 32 | 32 | 16 |
| Learning rate | 5e-5 | 2e-4 | 1e-4 |
| Optimizer | Adam | Adam | Adam |
| Scheduler | Cosine | Cosine | Cosine |
| Warm-up | 0.025 | 0.05 | 0.05 |
| Past frames | 6 | 6 | 1 |
| Rollout frames | - | 10 | 10 |

Table 8: Control task results of ours and ours with slot masks. The results of each model in seen tasks and blocks are presented with task success percentages based on the threshold distance of success. The results of each model in unseen tasks and blocks are presented with a threshold distance of 0.05.

| Model | Seen tasks & blocks | | | Unseen tasks | | | Unseen blocks |
|---|---|---|---|---|---|---|---|
| | 0.05 | 0.075 | 0.1 | T1 | T2 | T3 | |
| Ours | **50.0** | **61.5** | **73.0** | **24.5** | **16.0** | 26.5 | **38.0** |
| Ours + Mask | 46.5 | 59.0 | 70.0 | 25.0 | 15.0 | **29.0** | 34.0 |

former decoder instead of a transformer encoder in the action decoder, enabling fusion through cross-attention between the instruction and slot sequence. We perform these experiments for two grouping strategies: grouping by timestep and grouping by slot, with a threshold distance of 0.05 to measure success rates on seen tasks and blocks. The experiments are carried out with the number of decoder layers set to 2 and 8. The results show that when the layer count is 2, grouping by timestep achieves a success rate of 43.0%, and grouping by slot achieves 42.5%. As we increase the layer number to 8, the success rates decrease slightly to 37.5% for grouping by timestep and 40% for grouping by slot. This reaffirms that the injection of instructions with slots into the action decoder lowers the performance of action prediction.

A.3 ADDITIONAL VISUALIZATIONS

**Visualization of slots learned by SAVi and LSlotFormer** The top of Figure 5 provides qualitative examples of SAVi's performance when trained on our robotic dataset. We show that LSlotformer effectively predicts future video trajectories in the slot feature space that are conditioned on the given language instructions. The bottom of Figure 5 provides qualitative examples of LSlotformer's performance when trained on our robotic dataset, showing that the structure of the slots learned in SAVi is well-maintained after prediction.

**Baseline performance with various sampling steps** Figure 6 and 7 provides qualitative examples of Seer predicted trajectories with various DDIM sampling steps. The predictions of Seer-F does show image quality improvements when increasing the sampling step from 5 to 10 as we can see in top rows of 6. However, both Seer-F and Seer-S consistently fail to generate accurate predictions of arm movement and block locations regardless of how much sampling steps are increased if it is greater than 10. Results present limitations of Seer's performance despite increasing the DDIM sampling steps which implies that these models lack the mechanisms to effectively predict interactions in dynamic environments. Due to this result, we use 10 as a default diffusion sampling step for our experiments to efficiently run computation-heavy diffusion models without performance drop.

Figure 5: Qualitative visualization of slots learned by Slot Attention for Videos (SAVi) and our world model, LSlotFormer. In the top section, SAVi effectively segments scene frames into individual slots. In the bottom section, LSlotFormer uses language guidance to predict future states in slot form, with the decoded slots maintaining the structure learned by SAVi, showing consistency in representation.

**Additional evaluation of visuo-linguo-motor control tasks** Figure 8 and 9 provides qualitative examples of ours and baseline methods as visuo-linguo-motor controller for task evaluation in block-to-block language-table gym environment. As depicted in Figure 8, while Seer-F rarely shows meaningful behavior and Susie fails to accurately control target block to the destination, our method successfully execute instructions manipulating target blocks precisely to the objective. Depicted failure cases in 9 show that our method encounters specific limitations when target blocks are pushed beyond the boundaries of the board frame. In these cases, the models are difficult to generalize to identifying and controlling these out-of-bound blocks, since they rarely appear in expert dataset collected by tele-operation.

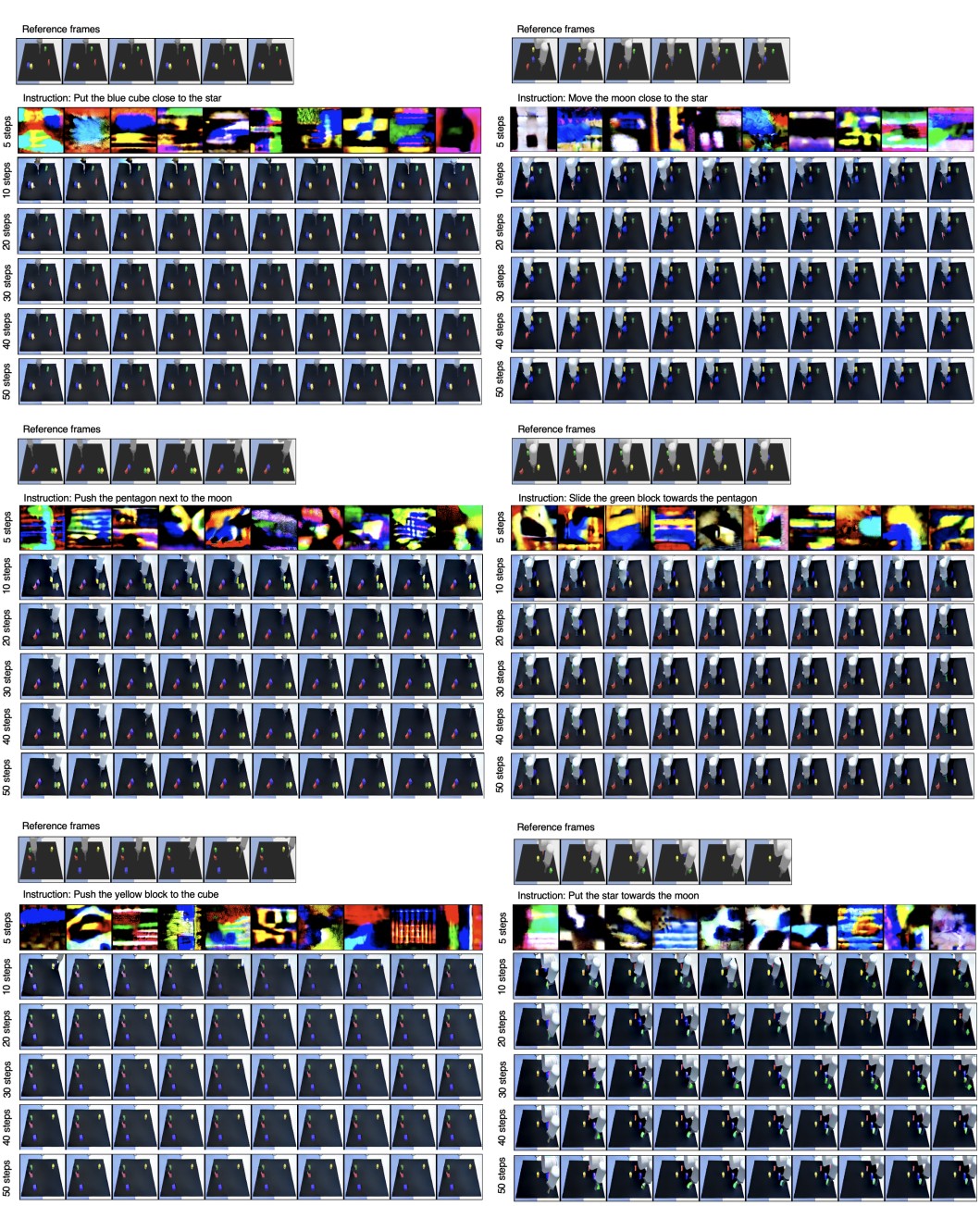

Figure 6: Decoded video frames of Seer-F with various DDIM sampling steps conditioned on given reference frames and the instruction. Regardless of sampling steps, Seer-F fails to accurately predict the arm movement and block locations.

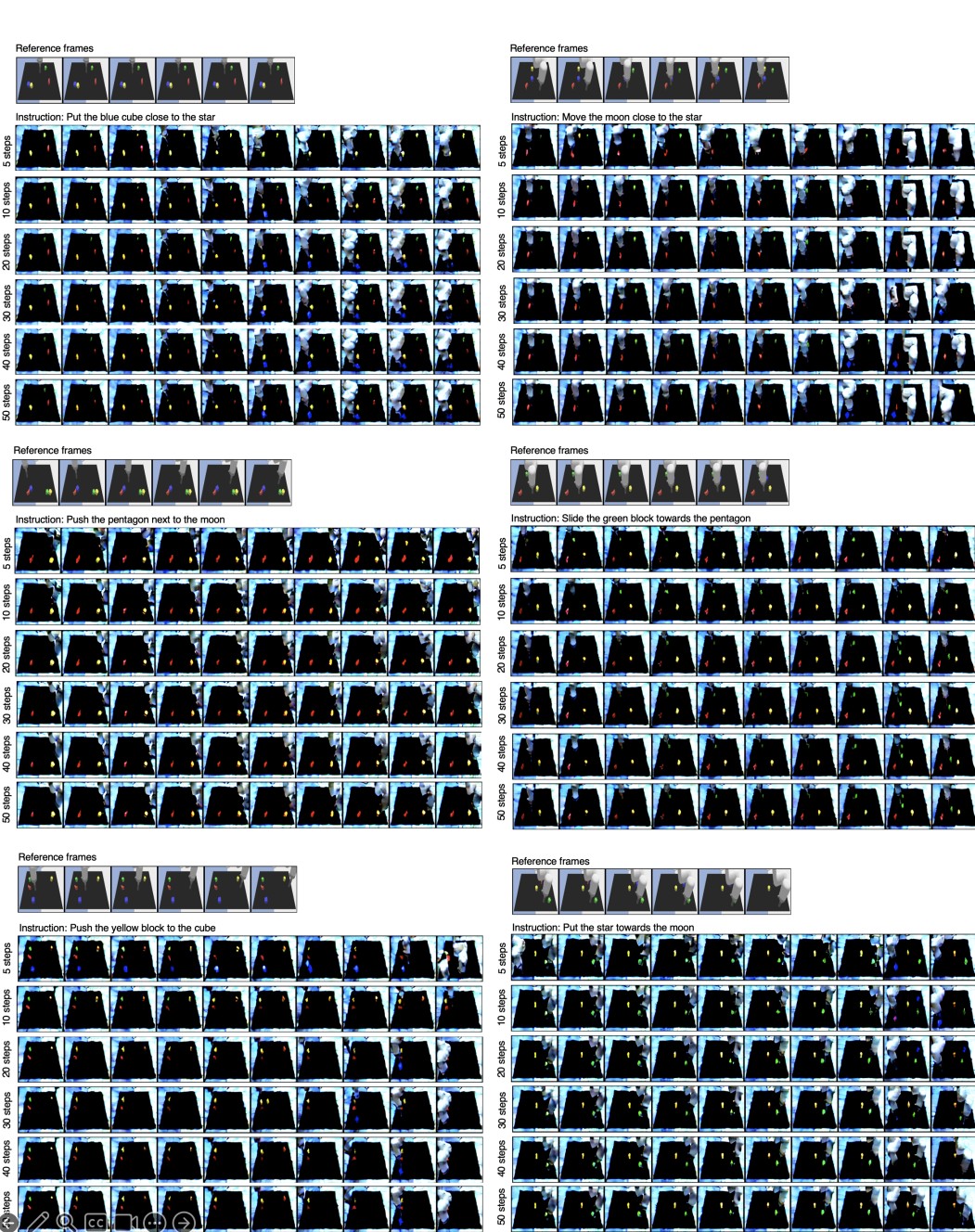

Figure 7: Decoded video frames of Seer-S with various DDIM sampling steps conditioned on given reference frames and the instruction. Regardless of sampling steps, Seer-S fails to accurately predict the arm movement and block locations.

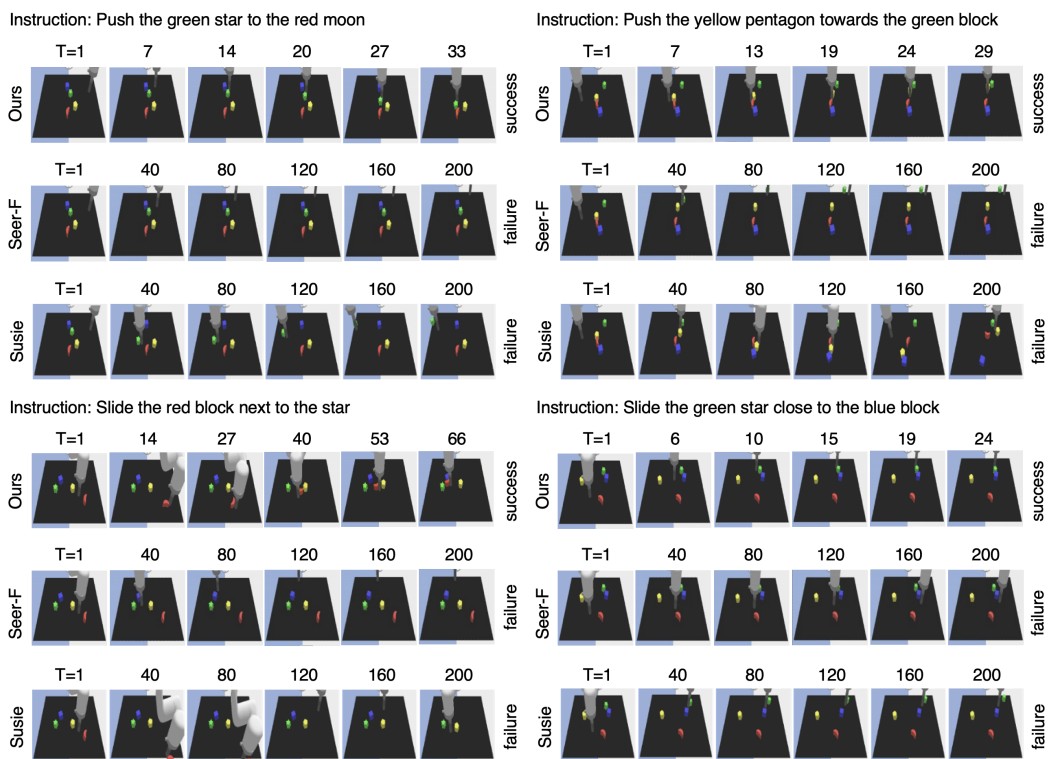

Figure 8: Additional predicted action trajectories of ours and the baselines in visuo-linguo-motor control simulation environment where our method succeeds but baselines fail.

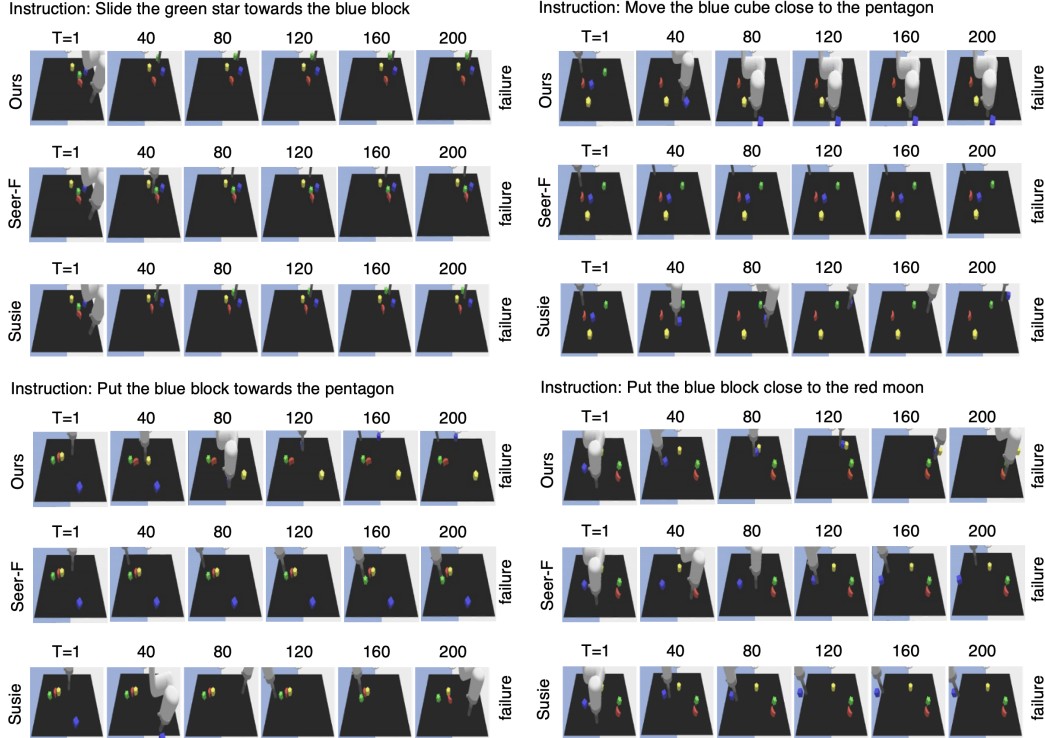

Figure 9: Additional predicted action trajectories of ours and the baselines in visuo-linguo-motor control simulation environment where our method and baselines fail.

