# OpenReview forum: "Language-Guided Object-Centric World Models for Predictive Control"
_ICLR.cc/2025/Conference — Submitted to ICLR 2025_

### Official Review · Reviewer_Eu5p · 2024-10-16

**Soundness:** 2
**Presentation:** 2
**Contribution:** 1
**Rating:** 3
**Confidence:** 3

**Summary:**

The authors propose to train a language-conditioned latent dynamics model whose state representations are object-centric “slots” provided by a frozen pre-trained model. They then train an inverse dynamics model that predicts the actions corresponding to the transitions of the autoregressively-generated latent slot representations.

**Strengths:**

- The paper is written quite clearly, I think the authors presented their ideas quite well
- I appreciate the ablation discussions in 4.6.

**Weaknesses:**

- Despite the paper being relatively clearly written, I would highly recommend using ICLR 2025’s extended page limit to increase the size and quality of your visualizations. For instance, for Figure 3, it’s very hard to see where the cube and moon are in the scene. I likewise cannot see any empirical quality differences between your approach’s generations and SuSiE’s.
- The approach is not tested on a wide range of tasks – only the simulated LanguageTable benchmark. It’s not at all clear to me that it would generalize to the real world. Given that SuSiE was evaluated both in sim (CALVIN) and on real-world robots on Bridge-like tasks and showed good performance (compared to strong real-world baselines like RT-2), it is unclear if the present paper’s approach would similarly scale to such more complex tasks.
- Similarly, the authors claim: “However, the major drawback of language-guided video-generation models is the requirement of large-scale labeled language-video datasets and the corresponding high computational cost. Therefore, latent predictive models, which abstract video to predict forward in compact latent state spaces, can serve as an alternative from a computational efficiency perspective.”
  - If this is true, it seems more sensible to evaluate in the real world, where data is more limited than in sim.
  - Additionally, SuSiE does show that an off-the-shelf image generation model pre-trained on general image-language data can be fine-tuned to work well with robot data just on existing relatively limited robot datasets. If that’s the case, it seems highly unclear that sample efficiency is a problem.
- “The task is to move a block to another block based on the description of the colors or shapes … which are the red moon, blue cube, green star, and yellow pentagon.” This likewise seems very limited – I understand that there are generalization experiments, but the Bridge dataset used for SuSiE’s real-world experiments contain a much wider range of actions and objects, and thus also a much wider range of language (including many noisy labels). It has thus demonstrated to be scalable to a wider range of language and visual entities, which I think would similarly benefit this approach (as it stands, being able to generate latent state trajectories for such a limited number of objects and actions does not say much about its scalability).
- As it stands, given that the approach was only evaluated on a single task setting and said setting is not that representative of real-world language-conditioned visuo-motor robotics tasks, I do not think that this approach has sufficiently demonstrated its general applicability. I think more experiments in a wider variety of domains would be very helpful, especially in real world experiments.
- Finally, I think it would be important to include results that showcase in what settings visual and object-centric world models each excel or break down. I can imagine some cases wherein image or video generation is bad: for example, if I ask a robot to fetch me something from a closed cabinet, the image generator would have to effectively “imagine” what the inside of that cabinet looks like. However, I do not have corresponding intuition for object-centered world models (though it seems like their weaknesses might be quite similar). See last question for more.

**Questions:**

- It’s interesting that Seer gets basically 0% on all tasks. What are the qualitative failure cases there?
- Since SuSiE was already evaluated on CALVIN, which is simulated, why not evaluate your approach in that setting?
- In what qualitative settings would object-centered world models be more or less effective than image ones? Do the authors have any intuitive examples of this, and is there any experimental evidence to back that up?

---

> ### Author Response · Authors · 2024-11-29
>
> Thanks for your valuable feedback. We are currently running additional experiments to address reviewers’ concerns, and below are our responses to the specific concerns if answerable without additional experimental results. We will fully cover all concerns during the extended review timeline, thanks for your understanding!
>
> > [Weakness 1] Despite the paper being relatively clearly written, I would highly recommend using ICLR 2025’s extended page limit to increase the size and quality of your visualizations. For instance, for Figure 3, it’s very hard to see where the cube and moon are in the scene. I likewise cannot see any empirical quality differences between your approach’s generations and SuSiE’s.
>
>  Thanks for pointing out the difficulty in parsing the qualitative results in the figures and visualizations, we improved figure visibility to make the results clearer and more interpretable. To aid understanding, we have added a video related to Figure 3 in the supplementary material. The video includes four segments: original.gif (the original video), ours.gif (the video decoded by our method), and seer_s.gif and seer_f.gif (videos generated by the Seer baselines). While Seer is restricted to generating 10 frames, our method, being autoregressive, has no such limitation. Additionally, as per your suggestion, we have increased the size of Figures 3 and 4 to improve readability.
>
>  However, both Seer and our method appear to lose shape details when decoding into the pixel space in Figure 3. In our case, this is likely due to the limitations of the image decoder, as well as the fixed resolution, which constrains the potential for quality improvement. We believe this limitation is acceptable, as the primary contribution of our method is not in producing high-quality frame decodings but in accurately predicting future states within the latent space.
>
>
> > [Weakness 2] The approach is not tested on a wide range of tasks – only the simulated LanguageTable benchmark. It’s not at all clear to me that it would generalize to the real world. Given that SuSiE was evaluated both in sim (CALVIN) and on real-world robots on Bridge-like tasks and showed good performance (compared to strong real-world baselines like RT-2), it is unclear if the present paper’s approach would similarly scale to such more complex tasks.
>
>  To show our method in a more challenging environment, we conducted additional experiments on more complex simulated environments (Libero) using our method with SAVi and with state-of-the-art video slot encoder model (SOLV). Please refer to the general comment for additional experiments.
>
>
> > [Weakness 3] Similarly, the authors claim: However, the major drawback of language-guided video-generation models is the requirement of large-scale labeled language-video datasets and the corresponding high computational cost. Therefore, latent predictive models, which abstract video to predict forward in compact latent state spaces, can serve as an alternative from a computational efficiency perspective.
>
> > - If this is true, it seems more sensible to evaluate in the real world, where data is more limited than in sim.
> > - Additionally, SuSiE does show that an off-the-shelf image generation model pre-trained on general image-language data can be fine-tuned to work well with robot data just on existing relatively limited robot datasets. If that’s the case, it seems highly unclear that sample efficiency is a problem.
>
>  We currently have no access to real world demonstration data and physical manipulation devices. Also, complex domain cases like real-world are limited due to our default setting’s (SAVi with simple CNN) expressiveness capability, which is addressed in additional experiments. Extending and testing in real-world scenarios is a priority for future work once the necessary resources become available.
>
> We considered internet scaled pre-training also as a part of the dataset and computation burden, which can be resolved leveraging an open-sourced model. However, the scale of limited fine-tuning data in their work is not very trivial compared to pre-training data scale. Also, our main contribution is computational efficiency during training and inference.

---

> > ### Author Response · Authors · 2024-11-29
> >
> > > [Weakness 4] “The task is to move a block to another block based on the description of the colors or shapes … which are the red moon, blue cube, green star, and yellow pentagon.” This likewise seems very limited – I understand that there are generalization experiments, but the Bridge dataset used for SuSiE’s real-world experiments contain a much wider range of actions and objects, and thus also a much wider range of language (including many noisy labels). It has thus demonstrated to be scalable to a wider range of language and visual entities, which I think would similarly benefit this approach (as it stands, being able to generate latent state trajectories for such a limited number of objects and actions does not say much about its scalability).
> >
> > > [Weakness 5] As it stands, given that the approach was only evaluated on a single task setting and said setting is not that representative of real-world language-conditioned visuo-motor robotics tasks, I do not think that this approach has sufficiently demonstrated its general applicability. I think more experiments in a wider variety of domains would be very helpful, especially in real world experiments.
> >
> > Regarding concerns about language diversity and scalability, we agree that the current experimental setup involves a relatively limited range of objects and language inputs and more comprehensive evaluation of our method's scalability is needed. To show our method in a more diverse environment, we conducted additional experiments on more complex simulated environments (Libero) using our method with SAVi and with state-of-the-art video slot encoder model (SOLV). Please refer to the general comment for additional experiments. Also, as stated in [Answer 3], we currently have no access to real world demonstration data and physical manipulation devices. Extending and testing in real-world scenarios is a priority for future work once the necessary resources become available.
> >
> >
> > > [Weakness 6] Finally, I think it would be important to include results that showcase in what settings visual and object-centric world models each excel or break down. I can imagine some cases wherein image or video generation is bad: for example, if I ask a robot to fetch me something from a closed cabinet, the image generator would have to effectively “imagine” what the inside of that cabinet looks like. However, I do not have corresponding intuition for object-centered world models (though it seems like their weaknesses might be quite similar). See last question for more.
> >
> > > [Question 3] In what qualitative settings would object-centered world models be more or less effective than image ones? Do the authors have any intuitive examples of this, and is there any experimental evidence to back that up?
> >
> > Since object-centric representations of individual entities are given as input, the world model explicitly learns relationships between objects such as physical interactions or compositional understanding. Prior works (Heravi et al., 2023; Yoon et al., 2023; Driess et al., 2023) and our experiments empirically show that using object-centric representation can benefit in robotics manipulation tasks compared to non-object-centric features. Also, Intuitively, object-centric world models are expected to excel in cases where there are multiple objects and especially when their motions are independent, for example, swarm robotics or pedestrian monitoring. Though this property is out of our work’s scope, it is a promising direction for future research.
> >
> > On the other hand, object-centric representations might be less effective in scenarios where their advantages are diminished, such as environments with very few objects or where one object dominates the scene, reducing the need for relational reasoning. In such cases, where no significant interactions occur, image-based models could perform better.
> >
> > Regarding the example of "imagination" such as a robot predicting the contents of a closed cabinet, we agree that object-centric world models face similar challenges. Their reliance on explicit object representations and interactions could limit performance in highly stochastic or ambiguous environments where generating plausible visualizations is more critical than understanding object relationships.

---

> > > ### Author Response · Authors · 2024-11-29
> > >
> > > > [Question 1] It’s interesting that Seer gets basically 0% on all tasks. What are the qualitative failure cases there?
> > >
> > >   In Figure 3 and Figure 6 and 7 in Appendix, Seer fails to predict accurate arm and object locations in the first hand. Though Seer fails to accurately recover image domains of language table environments, if it predicts arm and object information, the action decoder should be able to accurately predict actions. From this failed video prediction, in Figure 8 and 9 in Appendix, action predictions from Seer totally fail to solve the task with behavior of taking meaningless actions.
> > >
> > > > [Question 2] Since SuSiE was already evaluated on CALVIN, which is simulated, why not evaluate your approach in that setting?
> > >
> > >  To show our method in a more challenging environment, we conducted additional experiments on more complex simulated environments (Libero) using our method with SAVi and with state-of-the-art video slot encoder model (SOLV). Please refer to the general comment for additional experiments.
> > >
> > >
> > > Finally, we hope our response could address the concerns, and we thank the reviewer again for the helpful comments. We are glad to discuss further comments and suggestions.

---

### Official Review · Reviewer_vMaT · 2024-10-28

**Soundness:** 3
**Presentation:** 3
**Contribution:** 2
**Rating:** 5
**Confidence:** 4

**Summary:**

- The work proposes to inject language control in object centric world models and show its effectiveness in control.
- It argues that object centric models, specifically based on slots as studied in the paper, are more efficient and performant than large scale video generation models based on diffusion.
- They conduct experiments on a simulated table top manipulation benchmark to justify their method and various design choices.
- They present an analysis on how to tune these world models in terms of action decoding, look ahead steps, access to past states to achieve good performance.

**Strengths:**

- The paper is well written and easy to understand.
- The problem of building a world model for predictive control is a useful and relevant one to solve.
- The authors have ablated the components of their approach fairly well, including how to do the best action decoding, how many past steps to use in the world model etc.

**Weaknesses:**

- The authors do not have a SlotFormer baseline, which does not use any language conditioning.  Given that one of the key claims of the paper is that language conditioned object centric world models help downstream tasks, checking the importance of being language centric is critical. Adding that baseline would be helpful.
- For the evaluation of this approach, the authors have used the language table simulation environment, which involves some objects to be manipulated on a table top setting. This makes sense since there is a clear distinction between foreground (the objects) and background, which favors object centric approaches over general video generative models. However, showcasing some other scenarios or evaluation setups where maybe lets say intuitively a video generative model would have an edge, would have been interesting and more convincing to see.
- Minor: The qualitatives in figure 3 are not the easiest to parse, if the author’s method works well, but having a video to see the predictions would make the difference much clearer, I couldn’t find anything in the suppmat.

**Questions:**

A few questions (some overlapping with Cons. section above):
- Is language table the de-facto setting for studying object-centric control? It seems fairly limited and biased towards object centric approaches since it is clearly possible to discard the background information quite easily. Studying it in cases of ambiguity, where sometimes the background is obvious to ignore and sometimes not would bring more of the community to investigate this topic.
- In section 4.6, Is a world model really necessary? - Have the authors reported a pixel2action baseline, basically that does the same learning procedure, and except for learning from images directly, extract from some off the shelf network. The current results only ablate the absence of future slots, which makes sense, but that doesn't answer the question generally about needing a world model or not.

---

> ### Author Response · Authors · 2024-11-29
>
> Thanks for your valuable feedback. We are currently running additional experiments to address reviewers’ concerns, and below are our responses to the specific concerns if answerable without additional experimental results. We will fully cover all concerns during the extended review timeline, thanks for your understanding!
>
> > [Weakness 1] The authors do not have a SlotFormer baseline, which does not use any language conditioning. Given that one of the key claims of the paper is that language conditioned object centric world models help downstream tasks, checking the importance of being language centric is critical. Adding that baseline would be helpful.
>
> We conducted an additional experiment for the suggested SlotFormer baseline replacing LSlotFormer to SlotFormer with identical experiment settings only without language instruction. Slotformer prediction successfully reconstructed in-domain images, but semantically fails to perform block-to-block tasks. Action decoder trained with SlotFormer successed 2/200 (1%) of environment evaluation tasks.
>
> This is because in a language table environment, the agent does not have any information other than language instruction to perform given tasks. One can claim that even without instruction, when successfully learned, the model could be able to generate random A-to-B trajectory resulting in a success rate of 2/(n(n-1)); 2 for B-to-A execution leading to success. However, the predicted trajectories do not show a long-term task solving behavior due to its task ambiguity and inconsistency.
>
> > [Weakness 2] For the evaluation of this approach, the authors have used the language table simulation environment, which involves some objects to be manipulated on a table top setting. This makes sense since there is a clear distinction between foreground (the objects) and background, which favors object centric approaches over general video generative models. However, showcasing some other scenarios or evaluation setups where maybe lets say intuitively a video generative model would have an edge, would have been interesting and more convincing to see.
>
>  We expect video generative models, especially when pretrained, would have an edge in more complex image domains. To show our method in a more challenging environment, we conducted additional experiments on more complex simulated environments (Libero) using our method with SAVi and with state-of-the-art video slot encoder model (SOLV). Please refer to the general comment for additional experiments.
>
>
> > [Weakness 3] Minor: The qualitatives in figure 3 are not the easiest to parse, if the author’s method works well, but having a video to see the predictions would make the difference much clearer, I couldn’t find anything in the suppmat.
>
>  Thanks for pointing out the difficulty in parsing the qualitative results in the figures, we added video prediction visualizations in supplementary materials to make the results clearer and more interpretable.

---

> > ### Author Response · Authors · 2024-11-29
> >
> > > [Question 1] Is language table the de-facto setting for studying object-centric control? It seems fairly limited and biased towards object centric approaches since it is clearly possible to discard the background information quite easily. Studying it in cases of ambiguity, where sometimes the background is obvious to ignore and sometimes not would bring more of the community to investigate this topic.
> >
> > Since slot attention maps each input feature to a slot without exception, background is not processed differently or ignored, rather ideally, created as a background slot. Then, the world model learns to predict both objects and the background slot dynamics. The background slot can be ignored when training the action decoder as it learns to attend to necessary information for predicting the action.
> >
> > While the language table is in a less complex domain, it being our choice is done considering its multi-object property, availability of language instruction, offline data and evaluation environment and our default setting’s (SAVi with simple CNN) expressiveness capability. As we expected that our method can be applied when replacing image encoder with a more expressive network, we are currently running additional experiments to address this issue.
> >
> >
> > > [Question 2] In section 4.6, Is a world model really necessary? - Have the authors reported a pixel2action baseline, basically that does the same learning procedure, and except for learning from images directly, extract from some off the shelf network. The current results only ablate the absence of future slots, which makes sense, but that doesn't answer the question generally about needing a world model or not.
> >
> > To address your general question about whether a world model is necessary, we conducted an additional experiment. Specifically, we used the VAE encoder from Stable Diffusion v1.5 to extract features from the current frame and trained an action decoder using these features combined with the instruction. The results showed that only 1 out of 200 episodes succeeded, reinforcing our claim that future prediction is crucial for control tasks. For more details, please refer to [Answer 4] to Reviewer NMFa04.
> >
> > Finally, we hope our response could address the concerns, and we thank the reviewer again for the helpful comments. We are glad to discuss further comments and suggestions.

---

### Official Review · Reviewer_cCRo · 2024-10-31

**Soundness:** 2
**Presentation:** 3
**Contribution:** 2
**Rating:** 5
**Confidence:** 4

**Summary:**

The paper proposes to extend Slot Former to conditioned on language instruction object-centric dynamics prediction model. Such model could be used for decoding future actions for given state and instruction. Such predictions are in tern used for decoding the best action for the next time step. The paper showed that in the synthetic environment with large dataset, using such structured repression leads to better performance in comparison to using diffusion models for future state prediction. In addition, authors showed that such model is able to generalize to unseen tasks.

**Strengths:**

- The paper is well-written and mostly easy to follow.
- The authors provide a comparison with several image generation baselines adapted for the robotics domain, showing large gap from them.
- The authors study how robust their method to some changes in the environment, such as changing the block type or changing the task to unseen one.

**Weaknesses:**

- Overall, the proposed method is a simple modification of the SlotFormer adding language goal conditioned predictions and trained on a large dataset of the demonstrations.  On it its own it is not a big problem, if the proposed methods would be studied on diverse and challenging environments and compared with other methods that are state-of-the-art world models (e.g. [6, 7])

- While the improved performance on the synthetic dataset is encouraging, it is still not clear how the method would perform in more realistic scenarios where both object-centric models and corresponding agents can struggle. As mentioned by authors, recently it was shown that object-centric methods are able to decompose much challenging images or videos (e.g. see DINOSAUR (Seitzer et al. (2023)) for images or VideoSAUR [5] /SOLV (Aydemir et al., 2024)  for videos). Thus, it would be important to test how object-centric world models perform in more realistic environments with visual more complex scenarios, e.g. by training LSlotFormer on VideoSAUR or SOLV slots on environments like ManiSkill2).

- It is not clear how the methods compare to the standard baselines on this task: while outperforming diffusion models for video prediction, it is not clear if usage of world model with object-centric representations are comparable or not with state-of-the-art algorithms using the same data for training.

- Some experimental results would benefit from further analysis: for example, it not clear why using language conditioning for the agent itself is decreasing success rate.

- Some potentially missing related work in video-based object-centric learning, control with object-centric representation and world models based on the object-centric representations:

1. Focus: Object-centric world models for robotics manipulation  - also proposed a world model using object-centric representations. (https://arxiv.org/abs/2307.02427)
2. Learning Dynamic Attribute-factored World Models for Efficient Multi-object Reinforcement Learning, NeurIPS 2023 (https://arxiv.org/abs/2307.09205) - learns dynamics graph for more efficient policies.
3. Self-Supervised Visual Reinforcement Learning with Object-Centric Representations, ICLR 2020 - proposed a goal-conditioned transformer based policy (or action decoder in authors notation), https://arxiv.org/abs/2011.14381
4. Entity-Centric Reinforcement Learning for Object Manipulation from Pixels (https://arxiv.org/pdf/2404.01220)
5. Object-Centric Learning for Real-World Videos by Predicting Temporal Feature Similarities (extention of SAVi to more complex real-world videos using DINOSAUR), https://arxiv.org/abs/2306.04829


6. TD-MPC2: Scalable, Robust World Models for Continuous Control
7. PWM: Policy Learning with Large World Models

**Questions:**

- What is trained during "Predict future dynamics"  Figure 1(b)? If nothing remome "fire" sign near the world model?

- "nature of slot attention not being robust to variable object number" this could be clarified

---

> ### Author Response · Authors · 2024-11-29
>
> Thanks for your valuable feedback. We are currently running additional experiments to address reviewers’ concerns, and below are our responses to the specific concerns if answerable without additional experimental results. We will fully cover all concerns during the extended review timeline, thanks for your understanding!
>
> > [Weakness 1] Overall, the proposed method is a simple modification of the SlotFormer adding language goal conditioned predictions and trained on a large dataset of the demonstrations. On it its own it is not a big problem, if the proposed methods would be studied on diverse and challenging environments and compared with other methods that are state-of-the-art world models
>
> > [Weakness 3] It is not clear how the methods compare to the standard baselines on this task: while outperforming diffusion models for video prediction, it is not clear if usage of world model with object-centric representations are comparable or not with state-of-the-art algorithms using the same data for training.
>
>  To show our method in a more challenging environment, we conducted additional experiments on more complex simulated environments (Libero) using our method with SAVi and with state-of-the-art video slot encoder model (SOLV). Please refer to the general comment for additional experiments.
>
> For the baselines, we considered language conditioned diffusion video/image prediction models as world model baselines. Our motivation is to substitute diffusion world models with more efficient object-centric world models. Although direct comparison between these methods can be difficult due to their difference in design, our approach shows better performance over diffusion models, both when trained from scratch on the target dataset and when fine-tuned from a pre-trained model, while also offering greater computational efficiency.
>
> The primary motivation behind our research stems from studies such as UniSim (Yang et al., 2023) and UniPi (Du et al., 2023), which employ video diffusion models as world models but require substantial time and computational resources. For instance, UniPi utilizes 256 TPU-v4s to train video diffusion models for 2 million steps with a batch size of 2048. Similarly, UniSim uses 512 TPU-v3s to train video diffusion models for 1 million steps over 20 days, processing nearly 821 million data examples. Furthermore, these diffusion models require several sampling steps during inference, which can also pose significant challenges in practical applications.
>
> To address these challenges, we aimed to implement a world model using a latent predictive approach. Therefore, we compare our method against SOTA diffusion-based generative world models as baselines. While we greatly appreciate your suggestion to compare with other types of world models, we believe that doing so could divert the focus from the primary objectives of our work.
>
> Additionally, as shown in Table 2, we provide a comparison of the data usage between diffusion-based generative world models and our approach. Notably, we outperform Seer-S, a version of Seer trained from scratch using the same amount of data as our method. This result further underscores the efficiency and effectiveness of our proposed approach.
>
>
> > [Weakness 2] While the improved performance on the synthetic dataset is encouraging, it is still not clear how the method would perform in more realistic scenarios where both object-centric models and corresponding agents can struggle. As mentioned by authors, recently it was shown that object-centric methods are able to decompose much challenging images or videos (e.g. see DINOSAUR (Seitzer et al. (2023)) for images or VideoSAUR [5] /SOLV (Aydemir et al., 2024) for videos). Thus, it would be important to test how object-centric world models perform in more realistic environments with visual more complex scenarios, e.g. by training LSlotFormer on VideoSAUR or SOLV slots on environments like ManiSkill2).
>
>  To show our method in a more challenging environment, we conducted additional experiments on more complex simulated environments (Libero) using our method with SAVi and with state-of-the-art video slot encoder model (SOLV). Please refer to the general comment for additional experiments.

---

> > ### Author Response · Authors · 2024-11-29
> >
> > > [Weakness 4] Some experimental results would benefit from further analysis: for example, it not clear why using language conditioning for the agent itself is decreasing success rate.
> >
> >  As stated in “Does adding instructions into the action decoding help?” ablation, since the world model predicts a slot trajectory that already realizes given language instruction, the slot trajectory is sufficient for  predicting actions for the agent. While performance drop is minor (1~4%p) for transformer action decoders in Table 3, redundantly providing instruction information to the agent can rather hinder the training process.
> >
> > > [Weakness 5] Some potentially missing related work in video-based object-centric learning, control with object-centric representation and world models based on the object-centric representations
> >
> > Thanks for pointing out some missing related works, we checked and cited them.
> >
> > **Page 1:** Meanwhile, object-centric representation derived from Locatello et al. (2020) has recently garnered significant attention as a method for encoding images or videos in several studies (... Zadaianchuk et al., 2023).
> >
> > **Page 3:** A world model predicts future states based on current environments. Recent advancements in diffusion models have led to active research on video generation-based world models (Du et al., 2024a;b; Yang et al., 2024), but these methods demand substantial data and lengthy training and inference times. Some studies address these challenges by learning dynamics within the latent space, either by extracting full-image representations (Hafner et al., 2019; 2020; 2023; Babaeizadeh et al., 2017; Franceschi et al., 2020) or by using masked inputs (Seo et al., 2023). Others propose object-centric approaches, focusing on state representations (Wu et al., 2022; Collu et al., 2024) or combining states and actions (Ferraro et al., 2023; Feng & Magliacane, 2023). We introduce the first object-centric world model guided by natural language. Our model is more computationally efficient than diffusion-based approaches and outperforms non-object-centric methods in action prediction tasks. Unlike prior studies that train goal-conditioned policies using object-centric representations requiring goal images at test time (Zadaianchuk et al., 2020; Haramati et al., 2024), our approach predicts future states based on instructions and uses these predictions to train an action decoder, enabling manipulation tasks without goal images.
> >
> > —
> >
> > > [Question 1] What is trained during "Predict future dynamics" Figure 1(b)? If nothing remove "fire" sign near the world model?
> >
> >  The figure shows the autoregressive prediction process of the slot world model. Autoregressive prediction is done both in training and inference for long-term consistency of the world model, which means captions of the figure were misleading. We changed the captions to resolve this ambiguity.
> >
> >  **Page 2 Figure 1 caption:**  (b) More specifically, the world model utilizes the predicted slots from the previous steps to autoregressively predict future slots.
> >
> >
> > > [Question 2] "nature of slot attention not being robust to variable object number" this could be clarified
> >
> >  Slot attention needs a predefined number of slots near optimal to properly train the object discovery capability. Original slot attention literature shows experiments of being robust to increase in slot numbers at test time, however, this is when slot attention is already trained with a slot number of maximum object number + 1(background) which is optimal. A large slot number during training causes over-segmentation of objects harming its object discovery capability and robustness to numbers of objects. Since it would be common for a maximum number of objects to be not available (or has no upper bound) for manipulation tasks in reality unlike simulated benchmark environments, it is a critical limitation and promising direction for future research.
> >
> > Finally, we hope our response could address the concerns, and we thank the reviewer again for the helpful comments. We are glad to discuss further comments and suggestions.

---

### Official Review · Reviewer_NMFa · 2024-11-04

**Soundness:** 2
**Presentation:** 2
**Contribution:** 2
**Rating:** 3
**Confidence:** 4

**Summary:**

The paper introduces a language-guided, object-centric world model for predictive control, which is both computationally efficient and effective in robotic and autonomous tasks. Using slot attention for object-focused representation and language guidance, it outperforms diffusion-based models in task success, speed, and generalization.

**Strengths:**

- Effectively uses SAVi to extract object-centric frame features, enhancing computational efficiency and model accuracy.
- Compares against two baseline models (Seer and Susie), highlighting the advantages in efficiency and success rate of the proposed approach.
- Demonstrates generalization capabilities to unseen tasks and objects, showing robustness in diverse environments.

**Weaknesses:**

- The contribution in terms of "object-centric" design feels limited, as it primarily substitute the encoder for SAVi without introducing distinct object-centric innovations.
- The lack of an experiment comparing your proposed model with a VAE-based variant (ours + VAE in Tab.1) makes it difficult to conclusively justify the benefits of slot attention.
- Comparison against video diffusion models would be more appropriate than models like instructpix2pix, as diffusion models are more aligned with the proposed model's multi-frame prediction capability.
- The analysis suggesting that future state prediction alone suffices for action decoding is questionable; the low accuracy for "instruction + 0 future steps" (2.5%) compared to near-zero performance for Seer implies that baseline results may lack rigor, potentially outperforming when future states are not predicted.
- The dataset used is overly simplistic, limiting the scope of validation for the world model. Testing across multiple, varied environments would better demonstrate the model’s general applicability and robustness.

**Questions:**

See Weaknesses

---

> ### Author Response · Authors · 2024-11-29
>
> Thanks for your valuable feedback. We are currently running additional experiments to address reviewers’ concerns, and below are our responses to the specific concerns if answerable without additional experimental results. We will fully cover all concerns during the extended review timeline, thanks for your understanding!
>
>
> > [Weakness 1] The contribution in terms of "object-centric" design feels limited, as it primarily substitute the encoder for SAVi without introducing distinct object-centric innovations.
>
>  While most object-centric literatures are limited to image reconstruction and segmentation, integration to more practical downstream tasks can enrich the usability of object-centric representation. Therefore, rather than proposing a novel method of object-centric representation, we focus on properly utilizing object-centric representation which is done simply and efficiently using transformers with permutation invariance.
>
>
> > [Weakness 2] The lack of an experiment comparing your proposed model with a VAE-based variant (ours + VAE in Tab.1) makes it difficult to conclusively justify the benefits of slot attention.
>
>  The reason we did not explicitly include “ours + VAE” in the main text is that the primary purpose of our world model is not to decode into the RGB space but to predict states in the latent space of slots. We considered it somewhat unnatural to decode into the RGB space and then re-encode it using a VAE. However, your observation is valid, and we are currently conducting additional experiments to address this question. We will provide a response as soon as we have the results.
>
>
> > [Weakness 3] Comparison against video diffusion models would be more appropriate than models like instructpix2pix, as diffusion models are more aligned with the proposed model's multi-frame prediction capability.
>
>  One of the baseline models, Seer, is a language-conditioned video diffusion model conditioned on 6 frames to generate 10 future frames like our method.
>
> The motivation behind our research stems from studies such as UniSim (Yang et al., 2023) and UniPi (Du et al., 2023), which employ video diffusion models as world models but require substantial time and computational resources. For instance, UniPi utilizes 256 TPU-v4s to train video diffusion models for 2 million steps with a batch size of 2048. Similarly, UniSim uses 512 TPU-v3s to train video diffusion models for 1 million steps over 20 days, processing nearly 821 million data examples. Furthermore, these diffusion models require several sampling steps during inference, which can also become a significant obstacle in practical applications.
>
> To address these challenges, we aimed to implement a world model using a latent predictive approach. To evaluate the time efficiency of our method, we selected Seer (Gu et al., 2023), a latent video diffusion model known for its strength in time efficiency, as a baseline. As you pointed out, SuSiE (Black et al., 2023) may not align perfectly in terms of multi-frame prediction capability. However, if we consider its functionality solely as a world model, it can be seen as more efficient than video diffusion models like Seer due to its ability to generate only the essential frames. Additionally, because it generates frames autoregressively, it tends to produce coarser videos. From this perspective, we concluded that SuSiE, based on instruct2pix2pix, is a suitable comparison point for time efficiency as a generative world model. Its strong time efficiency aligns well with our goal of benchmarking the proposed model against generative methods in this aspect.

---

> ### Author Response · Authors · 2024-11-29
>
> > [Weakness 4] The analysis suggesting that future state prediction alone suffices for action decoding is questionable; the low accuracy for "instruction + 0 future steps" (2.5%) compared to near-zero performance for Seer implies that baseline results may lack rigor, potentially outperforming when future states are not predicted.
>
>  Your observation that it might be possible for baseline methods to achieve success using only the current frame and instruction, without future-predicted slots, is valid, especially since our proposed method demonstrated around a 2.5% success rate under these conditions. To test this hypothesis, we trained an action decoder (policy) using features extracted by inputting the current RGB frame into a VAE, along with the instruction. We evaluated this approach in the same language table setting described in the paper across 200 episodes. The resulting success rate was only 0.5%, supporting our claim that control tasks are challenging without future prediction, relying solely on the current state and instruction.
>
> Your implication that the Seer results with future prediction may lack rigor is also reasonable. However, as presented in the main text and supplementary material (Figure 3, Figure 6-7), this is likely due to Seer’s consistent failure in future prediction.
>
> > [Weakness 5] The dataset used is overly simplistic, limiting the scope of validation for the world model. Testing across multiple, varied environments would better demonstrate the model’s general applicability and robustness.
>
>  To show our method in a more challenging environment, we conducted additional experiments on more complex simulated environments (Libero) using our method with SAVi and with state-of-the-art video slot encoder model (SOLV). Please refer to the general comment for additional experiments.
>
> Finally, we hope our response could address the concerns, and we thank the reviewer again for the helpful comments. We are glad to discuss further comments and suggestions.

---

### Author Response · Authors · 2024-11-29
**General Response**

Thanks for all the reviewers' valuable feedback. The main concerns for our work that all reviewers had is that our experiments are conducted in a limited and simple environment setting, lacking diversity and scalability to more complex environments. We are aware that the language table environment is not sufficiently complex to represent the model’s general applicability, and would like to address this in the general comment.

Original SAVi with simple CNN image encoder can be applied to complex simulated image domain (MOVi++) with conditioning slot priors on segmentations or bounding boxes and training with optical flow objective from additional prediction network (PWC-Net). As stated in Appendix A.1, we stick to the non-conditional slot prior and reconstruction loss setting of SAVi for its simplicity and fully self-supervised pipeline without additional networks, which limits slot encoder’s applicability in complex domains.

We have tried using our slot encoder setting(SAVi with simple CNN) to train in Libero dataset and environment, however, expressiveness capabilities for this slot encoder setting couldn’t able to properly bind objects in complex domains. In theory, since our method is applicable with any slot encoder other than SAVi, we conducted additional experiments on more complex simulated environments (Libero) using our method with SAVi and with state-of-the-art video slot encoder model (SOLV).

During the rebuttal period, we attempted to extend our methodology by applying SOLV to Libero dataset. However, it has been challenging to complete this within the given period. Unfortunately, our current experiments have not yet demonstrated satisfactory performance when applying SOLV to Libero. Visualizations of the execution results reveal that the robotic arm can identify and move the objects necessary to complete the episode. However, the lack of precision in the actions seems to prevent successful episode completion.

This issue likely stems from the difference in action complexity between the two environments. In the language table setting, actions are represented as simple 2D displacements in the x and y directions, whereas Libero requires more complex 7D actions, including displacements in x, y, z directions, rotation, and the state of the gripper.

We explored two approaches for training SOLV: training from scratch using only Libero and fine-tuning from the checkpoint trained on the YouTube-VIS 2019 dataset (Yang et al., 2019) provided by the authors. While fine-tuning appeared to improve object segmentation, it did not have a significant impact on the success rate during evaluation.

Additionally, we investigated replacing the Invariant Slot Attention used in SOLV with standard Slot Attention, considering that Invariant Slot Attention might exclude object-related information such as position and size from the slot representation, which might be crucial for manipulation tasks. However, this modification also did not improve episode success.

Observing that the world model trained with SOLV slots fine-tuned using Invariant Slot Attention achieved a significantly lower loss while the action decoder’s loss increased, we hypothesized that the slots lacked critical information for manipulation tasks such as object position, size, and rotation. To address this, we adjusted the training of the world model to also predict relative grid information that reflects object position, size, and rotation. Simultaneously, the action decoder was trained to use both the slots and the relative grid information to predict actions. While this approach slightly increased the world model’s loss, it resulted in improved action decoder performance. Unfortunately, this improvement did not translate into higher episode success rates.

Based on these experiments, we conclude that the primary reason for failure during evaluation lies not in SOLV or the world model but in the action decoder’s inability to generate sufficiently precise actions. Further refinement of the action decoder is likely needed to overcome this limitation. Nevertheless, as our method is compatible with different action decoders, we believe this issue can be addressed by introducing a more sophisticated action decoder in the future. This would enable our method to handle the increased complexity of Libero’s action space effectively.

---

### Meta-Review · Area_Chair_GUhp · 2024-12-21

**Metareview:**

The paper introduces a language-guided, object-centric world model for predictive control, which is both computationally efficient and effective in robotic and autonomous tasks. Using slot attention for object-focused representation and language guidance, it outperforms diffusion-based models in task success, speed, and generalization.

While the paper tackles an interesting problem with some promising results, reviewers had a number of concerns regarding the novelty and contribution of the approach as well as the baselines considered.

**Additional Comments On Reviewer Discussion:**

The reviewers and authors had a discussion but all reviewers remained negative and had remaining concerns at the end of the rebuttal period.

---

### Decision · Program_Chairs · 2025-01-22

Reject